# New Insights on Glutathione’s Supramolecular Arrangement and Its In Silico Analysis as an Angiotensin-Converting Enzyme Inhibitor

**DOI:** 10.3390/molecules27227958

**Published:** 2022-11-17

**Authors:** Antônio S. N. Aguiar, Igor D. Borges, Leonardo L. Borges, Lucas D. Dias, Ademir J. Camargo, Pál Perjesi, Hamilton B. Napolitano

**Affiliations:** 1Grupo de Química Teórica e Estrutural de Anápolis, Universidade Estadual de Goiás, Anapolis 75132-903, GO, Brazil; 2Centro de Pesquisa e Eficiência Energética, CAOA Montadora de Veículos LTDA, Anapolis 75184-000, GO, Brazil; 3Escola de Ciências Médicas e da Vida, Pontifícia Universidade Católica de Goiás, Goiania 74605-010, GO, Brazil; 4Laboratório de Novos Materiais, Universidade Evangélica de Goiás, Anapolis 75083-515, GO, Brazil

**Keywords:** glutathione, supramolecular arrangement, M06-2X/6-311++G(d,p), DFT, molecular docking, angiotensin-converting enzyme

## Abstract

Angiotensin-converting enzyme (ACE) inhibitors are one of the most active classes for cardiovascular diseases and hypertension treatment. In this regard, developing active and non-toxic ACE inhibitors is still a continuous challenge. Furthermore, the literature survey shows that oxidative stress plays a significant role in the development of hypertension. Herein, glutathione’s molecular structure and supramolecular arrangements are evaluated as a potential ACE inhibitor. The tripeptide molecular modeling by density functional theory, the electronic structure by the frontier molecular orbitals, and the molecular electrostatic potential map to understand the biochemical processes inside the cell were analyzed. The supramolecular arrangements were studied by Hirshfeld surfaces, quantum theory of atoms in molecules, and natural bond orbital analyses. They showed distinct patterns of intermolecular interactions in each polymorph, as well as distinct stabilizations of these. Additionally, the molecular docking study presented the interactions between the active site residues of the ACE and glutathione via seven hydrogen bonds. The pharmacophore design indicated that the hydrogen bond acceptors are necessary for the interaction of this ligand with the binding site. The results provide useful information for the development of GSH analogs with higher ACE inhibitor activity.

## 1. Introduction

Cardiovascular disease (CD) is still the leading cause of mortality worldwide [1]. It is one of the costliest diseases for governments and healthcare systems (up to USD 320 billion/year) [2]. In 2019, according to a report from the World Health Organization (WHO), 17.9 million people died from CD, including clinical disorders of the heart and blood vessels [3]. There are several risk factors for CD, such as high blood pressure, diabetes mellitus, smoking, dyslipidemia, and being overweight/obesity [4,5]. Regarding increased blood pressure, its physiopathology is multifactorial and based on salt intake, obesity/insulin resistance, the influence of the sympathetic nervous system, and the renin–angiotensin–aldosterone system [6]. Among these factors, the renin–angiotensin–aldosterone system exerts a controlling function on blood pressure and hypertension via the synthesis of angiotensin II (a potent vasoconstrictor) [7] catalyzed by the angiotensin-converting enzyme (ACE, EC 3.4.15.1) [8]. ACE inhibitors comprise the first line of medicines employed in hypertension therapy, heart failure, myocardial infarction, and diabetic nephropathy [9].

The history of the development of inhibitors of the enzyme started in the 1960s when Rocha e Silva recognized that the effect of bradykinin (an important vasodilator peptide in controlling blood pressure) could be boosted by some peptides (BFP) found in the venom of a Brazilian snake, *Bothrops jararaca* [10]. Later, it was recognized that these peptides could inhibit the enzyme (named kininase II at that time) that can reduce the vasodilation effect of bradykinin. Further research clarified that the molecular basis of the enzymatic inhibition of bradykinin inactivation is the same as that in the formation of angiotensin II from the inactive angiotensin I. Currently, the enzyme catalyzing these reactions is called ACE [11]. Based on these observations, a series of compounds have been synthesized [12] and isolated from natural sources [13,14] that showed an effective inhibitory action on the enzyme [9]. Today, ACE inhibitors are widely used to manage hypertension [15]. Captopril, enalapril, and lisinopril are commercially available antihypertensive drugs to treat hypertension. However, their use presents some adverse effects, for instance, first-dose hypotension, hyperkalemia, renal dysfunction, angioedema, and cough [16,17].

In this regard, there is a great interest in developing new ACE inhibitors that present lower adverse effects compared to these clinically used antihypertensive drugs. As selected examples, some candidates are evaluated and described in the literature, such as the analogs of lisinopril [18], thymosin alpha-1 (Thα1) peptide [19], gonadotropin-releasing hormone (GnRH) [20], C-domain-specific phosphinic inhibitor, RXPA380 [21], and glucosides derived from eugenol [22]. Some of these studies also reported the characterization and modeling by molecular docking of the interaction between the candidates and the active site residues of the ACE.

Furthermore, the literature survey indicated that oxidative stress is frequently associated with high blood pressure [23]. Based on these clinical observations, several compounds with antioxidant capacity (among them GSH) were tested as potential ACE inhibitors. Such compounds (similar to captopril, a registered ACE inhibitor with thiol functionality) could act as dual-acting antihypertensive agents. In vitro inhibitory studies showed GSH to have ACE inhibitory activity falling into the μM range [23,24]. For example, the K_i_ constants for GSH and lisinopril against an angiotensin-converting enzyme purified from human plasma were determined as 11.7 μM and 0.662 nM, respectively [23].

Glutathione is a tripeptide composed of glutamine, cysteine, and glycine, which plays a pivotal role as a biological antioxidant [25,26]. Moreover, it is widely applied in the cosmetic, food, and pharmaceutical industries [27], as described in the European Pharmacopoeia [28]. In this regard, the supramolecular arrangements of GSH polymorphs (GSHA and GSHB) and molecular modeling by density functional theory (DFT) were compared to verify the conformations of the lowest energies in the system. The electronic structure by the frontier molecular orbitals and the molecular electrostatic potential (MEP) map were carried out to understand the molecular contact regions and predict the regions of electrophilic and radical attacks during the biochemical processes. In addition, the hydrogen bonds on supramolecular arrangements were studied by Hirshfeld surfaces (HS), quantum theory of atoms in molecules (QTAIM), and natural bonding orbital (NBO) analyses. Finally, in silico analysis validated the co-crystallized ligand (lisinopril, a commercial ACE inhibitor) with the angiotensin-converting enzyme via redocking analysis. To evaluate the GSH ACE inhibitor, the same model employed in the lisinopril redocking was also used for the GSH in the ACE.

## 2. Results and Discussion

### 2.1. Solid State Analysis

The crystal structure of the GSH is a ubiquitous thiol-containing tripeptide (*L*-γ-glutamyl-*L*-cysteinylglycine) that exists in its zwitterionic form. This structure is also reflected in the IR spectrum of the compounds recorded in KBr. Above the 3000 cm^−1^ regions, several associated OH, NH, NH_2_, and NH_3_^+^ (3346, 3249, and 3126) bands can be seen. A band at 2524 cm^−1^ can be associated with the cysteine thiol (SH) group. The 1713 cm^−1^ band corresponds to the νC=O band of the protonated carboxyl groups, while the strong bands at 1537 and 1385 cm^−1^ are related to the asymmetric and symmetric stretches, respectively, of the deprotonated CO_2_^–^ group [29].

The low molecular weight sulfur (thiols) present in GSH are easily oxidized and can be regenerated rapidly; these characteristics allow them to play an essential role in cell biology, such as protecting cells via an antioxidant process [30]. The polymorphism has been observed only under ambient pressure conditions, revealing that their polymorphs crystallize in the orthorhombic space group P2_1_2_1_2_1_. The differences in the geometry, crystal data, and structure refinement details for GSHA and GSHB are summarized in Table 1. The ORTEP diagrams of the GSHA and GSHB can be found in Figure 1, as well as the overlap of these polymorphs.

The results obtained for the GSH conformer molecular geometries were compared to the crystallographic model through the mean absolute deviation percentage formula,
(1)$$\mathrm{MADP}=\frac{100}{n}\overset{n}{\underset{i=1}{\sum }}\left|\frac{\chi_{DFT}-\chi_{XRD}}{\chi_{XRD}}\right|,$$
where $\chi_{DFT}$ and $\chi_{XRD}$ indicate the theoretical and experimental bond length or angle, respectively. The exchange and correlation functional M06-2X captures medium-range electronic correlation and can satisfactorily describe scattering interactions such as non-covalent interactions [31,32]. Other functionals were tested, such as the hybrid functional B3LYP, the highly parameterized empirical functional M06-HF [31], and the double-hybrid functional B2-PLYP [33]. However, the MAPD values showed that the functional M06-2X can satisfactorily describe scattering interactions, such as non-covalent interactions [31]. The graph presented in Figure 2 shows the results of the MAPD values obtained for the different levels of theory for GSH.

The MAPD values obtained for bond lengths and angles, at the M06-2X/6-311++G(d,p) level of theory, were 1.994 and 1.530%, respectively, where the Pearson correlation coefficients were 0.9790 and 0.9816. Figure 3 presents the graphs comparing the geometric properties obtained experimentally and theoretically. Regarding the lengths, stretching of the N_2_–C_4_, N_3_–C_8_, O_1_–C_5_, O_6_–C_10_, and C_10_–C_9_ and the compression of the N_1_–C_1_ and O_2_–C_5_ bonds were observed. In the GSH crystal, the molecules are found in the zwitterionic form, observed in the glutamic acid portion. In this structure, the O_1_–C_5_ and O_2_–C_5_ bonds have similar lengths (1.24 and 1.25 Å, respectively), making clear the resonant effect stabilizing the zwitterion conjugate base. However, theoretical calculations show that the isolated proton remains connected to the O_1_ atom, stretching the O_1_–C_5_ bond by 6.4% and compressing the O_2_–C_5_ bond by 4.9%.

The transit of the proton from the primary amino group to the conjugate base also compresses the N_1_–C_1_ bond by 2.8%, leaving the electron pair of the N atom free to resonate along the bond. The calculations also showed that the N_2_–C_4_ and N_3_–C_8_ peptide bonds are stretched by 3.2 and 3.5% in the free form of the molecule. Finally, the O_6_–C_10_ bond of the glycine portion is stretched by 2.6% in the isolated form of the molecule. This effect is explained by the strong O_6_–H⋯O_3_ interaction in the crystalline state, causing the proton to approach the O_6_ atom, stretching O_6_–C_10_.

The zwitterionic form of GSH is predominant in the crystalline state. However, thermodynamic calculations (Table 2) showed that, when it is kept isolated, the lowest energy state for the molecule is the neutral form, and 134.03 kcal/mol is more stable than the ionized form. In addition, all thermodynamic parameters (internal energy, enthalpy, free energy, entropy, etc.) calculated for both forms resulted in values lower than their ionized state, except in the case of entropy, which is 18.47 cal/mol·K higher for the neutral form. A relaxed scan calculation was carried out to verify the change in the total energy of the molecular system in which the proton starts from the N_1_ atom, bound at 1.0 Å, towards the O_1_ atom. From the scan plot shown in Figure 4, it is possible to verify that the system’s total energy decays as H^+^ approaches the conjugated carbonyl base.

Deviations greater than 2.0% also occurred in the bond angles. In the carboxyl group of the glutamic acid portion, a 4.3% increase in the O_2_–C_5_–C_1_ angle and a 2.3% decrease in the O_1_–C_5_–O_2_ angle were observed; the angles formed by C_1_ were also increased by the proton transition in the molecular structure (3.2% in N_1_–C_1_–C_2_ and 3.4% in N_1_–C_1_–C_5_). On the other hand, the C_5_–C_1_–C_2_ angle decreased by 5.2%. However, the distinction between the molecular structures of GSH in both polymorphs lies in the torsion of the tripeptide’s carboxyl group of the glycine portion. The overlap of the peptide bond, O_4_=C_8_–N_3_–C_9_, showed that the dihedral angles are −6.93° and 1.23°, respectively, in GSHA and GSHB, and the –COOH groups are at 73.63° and 77°.51° out of the plane of the peptide bond in the respective polymorphs. Figure 5 shows the overlapping of the molecular structures of GSHA and GSHB, starting from this dihedral, where the RMS was 0.0292 Å. Torsions in the other portions of the molecule were also observed. Namely, the dihedral angle of C_8_–C_6_–C_7_–S (cysteine portion) is −66.29° and −58.05°, respectively, while the dihedral angle of the glutamic acid portion is 145.94° and 112.44°.

### 2.2. Molecular Modeling Analysis

The frontier molecular orbitals, HOMO and LUMO, are represented in Figure 6, and their values are shown in Table 3. These orbitals resulted in a very high energy gap (242.6 kcal/mol), calculated by $\mathrm{GAP}=E_{\mathrm{LUMO}}-E_{\mathrm{HOMO}}$, and indicating the high kinetic stability of the GSH molecule. Its antioxidant power is related to protection against reactive oxygen species and electrophilic species produced in cellular oxidative processes; therefore, the values found for the energies of the frontier orbitals agree with this statement, indicating that the GSH molecule is a reductant ($E_{\mathrm{HOMO}}<0$), but not an oxidant ($E_{\mathrm{LUMO}}>0$). HOMO has a lone pair and pure *p* orbital characteristics, whose occupancy is 1.85558*e*. LUMO, on the other hand, is an $\sigma^{*}$ antibonding orbital located on the longitudinal axis of the C_10_–O_5_ bond, formed by the contribution of 65.02% of the C_10_ atom *sp*^1.86^ and 34.98% of the O_5_ atom *sp*^1.37^; it has an occupancy of 0.01811*e*.

The sulfhydryl group (–SH) of the cysteine portion is highly polarizable, characterizing it as a good nucleophile [34,35]. The σ bonding orbital of the S-H bond has an occupancy of 1.99027*e* and is formed by the contribution of 57.66% of the *sp*^5.61^*d*^0.04^ orbital of S and 42.34% of the *s* orbital of H (Figure 7a). The lone pair form the $\eta_{1}$ bonding orbitals (*sp*^0,47^), with an occupancy of 1.98974*e* (Figure 7b), and $\eta_{2}$ bonding orbital (*p*), with an occupancy of 1.95353*e* (Figure 7c). The calculated energies of these orbitals were −406.650, −460.529, and −213.606 kcal/mol, explaining the process of oxidation of GSH to glutathione disulfide through the activity of the glutathione oxidase enzyme, glutathione peroxidase, and glutathione reductase.

From the energies of the HOMO and LUMO, the chemical reactivity descriptors of the GSH were determined as the potential chemical,
(2)$$\mu ={\left(\frac{\partial E}{\partial N}\right)}_{\upsilon }=-\frac{I+A}{2}=-\chi ,$$
chemical hardness
(3)$$\eta ={\frac{1}{2}\left(\frac{\partial^{2}E}{\partial N^{2}}\right)}_{\upsilon }=\frac{I-A}{2},$$ and the global electrophilicity index,
(4)$$\omega =\frac{\mu^{2}}{2\eta }$$
where E is the energy of the system, N is the number of electrons, υ is the external potential generated by nuclei, $I\approx -E_{\mathrm{HOMO}}$ is the ionization potential, and $A\approx -E_{\mathrm{LUMO}}$ is the electron affinity. The high values found for the energy gap, as well as η, indicate that the GSH molecule is kinetically stable, having a low electron affinity and high electron transfer power during chemical processes.

Compared to other organic compounds, the results show that the GSH molecule has a nucleophilic character (ω< 0.93 a.u.) [36]. Oxygen atoms of the carboxyl groups and the carbonyl groups of GSH have a high charge density, showing the behavior of a Lewis base. This can be seen by the red color on the electronic isodensity surface of the MEP map represented in Figure 8. On the other hand, regions of lower charge density appear in blue and show the behavior of a Lewis acid.

To determine the local electrophilicity, the Fukui function [37,38] was used to predict the regions of nucleophilic,
(5)$$f^{+}={\left[\frac{\partial \rho \left(r\right)}{\partial N}\right]}_{\upsilon \left(r\right)}^{+},$$
electrophilic,
(6)$$f^{-}={\left[\frac{\partial \rho \left(r\right)}{\partial N}\right]}_{\upsilon \left(r\right)}^{-},$$
and radical attacks,
(7)$$f^{0}={\left[\frac{\partial \rho \left(r\right)}{\partial N}\right]}_{\upsilon \left(r\right)}^{0}.$$

Radical reactions caused by pathological processes, by the administration of oxidizing drugs, or even by physical activities can cause oxidative transformations of phospholipids, proteins, and deoxyribonucleic acid (DNA) in cell membranes. According to Fukui index calculations, the O_1_ and O_2_ atoms of the carboxyl group and the N_1_ atom of the amine group in the glutamic acid portion (Figure 9a) are favorable to the capture of free radicals produced in these processes, which explains the antioxidant character of GSH. Furthermore, the S atom in the cysteine portion can also undergo radical attacks resulting in the oxidized form of GSH, glutathione disulfide (GSSG), by the intervention of the enzymes glutathione oxidase or glutathione peroxidase. The calculations of the Fukui indices also indicated that GSH has active sites favorable to electrophilic attacks, identified by the isosurface regions in Figure 9b. All the oxygen and nitrogen atoms along the GSH molecule favor this type of attack. This can be explained by the mesomeric effect caused by the delocalized electrons of the carboxyl and amide groups in the tripeptide molecule. In addition, C atoms are favorable to electrophilic attacks, resulting from the inductive effect caused by the presence of O and N atoms, which reduces the charge density in the carbon chain skeleton.

### 2.3. Supramolecular Arrangement

The molecular topology of the crystals by HS mapped with $d_{norm}$ helps us to understand better the crystallographic forces driving the molecular arrangement and the interactions reported in Table 4. The contacts that are shorter than the sum of the van der Waals radii are represented by the red spots on a predominantly blue surface [39]. The crystal packing of GSHA is stabilized by the bifurcated intermolecular interactions S–H⋯O_1_ and S–H⋯O_2_ observed in the cysteine portion, involving the thiol group forming a motif [40] $R_{1}^{2}\left(4\right),$ and, for GSHB, S–H⋯O_1_ was observed forming a motif $C_{1}^{1}\left(11\right)$ and the intermolecular interactions N_1_–H⋯O_2_ [$C_{1}^{1}\left(5\right)$] and N_2_–H⋯O_2_ [$C_{1}^{1}\left(8\right)$]; these contacts form bifurcates in each compound. The red convex area above the glutamic acid correlates with the hydrogen bond interactions N_1_–H⋯O_4_ [$C_{1}^{1}\left(5\right)$] and N_1_–H⋯O_1_ [$C_{1}^{1}\left(5\right)$]. Furthermore, the interactions O_6_–H⋯O_3_ [$C_{1}^{1}\left(10\right)$] and N_3_–H⋯O_5_ [$C_{1}^{1}\left(5\right)$] involve part of the glycine. The presence of weak non-classical H-bond interactions is also evident in the GSHA, which is C_2_–H⋯O_4_ [$C_{1}^{1}\left(8\right)$] and, for GSHB, C_3_–H⋯O_2_ [$C_{1}^{1}\left(6\right)$].

The fingerprint plots (Figure 10) allow us to analyze the differences in the intermolecular patterns of the contacts and quantitatively evaluate the contributions among atoms [41]. The contacts involving H⋯O accumulated a percentage of 50% and are viewed as a distinct pair of spikes, evidence that the H-bonds are dominant in the crystalline environment [42]. The H⋯H contacts, close to 34%, with a decrease of 0.8% for GSHB, contribute to the overall crystal packing. In contrast, the proportion of S⋯H increased from 6.3% for GSHA to 9.6% for GSHB. In addition, weak C⋯H contacts decreased from 3% in GSHA to 2% in GSHB, indicating that the GSHA forms more H-bonds than GSHB.

The topological parameters for the intermolecular interactions in the molecular arrangements of the GSH polymorphs are presented in Table 4, and the molecular graphs are represented in Figure 11. In QTAIM, the observable properties of the molecular system are contained in electron density ρ(r), in which the Laplacian electron density, $\nabla^{2}\rho$, determines the depletions or peaks of the electron charge concentration between nuclear attractors, indicating the location of bond critical points (BCPs) [43,44,45]. In other words, $\nabla^{2}\rho$ corresponds to the concentration of the electronic charge in the intranuclear region of two attractors. If the electron density is accumulated in the intranuclear region, its value is negative in the BCP, and the interaction is shared so that the attractors are covalently attached. On the other hand, if $\nabla^{2}\rho >0$, the electron density is concentrated in the attractors, a closed-shell interaction in which the attractors are linked by weak interactions. The low values of the electron density (ρ<0.1) and the positive Laplacian found on the BCP indicate closed-shell interactions [43,44,45,46]. Furthermore, when the values for the ratio |v|/G<1.0 and the total energies are very low, h(r)≈0, this indicates the intermolecular interactions in the supramolecular arrangement of GSH are of low intensity, configuring hydrogen bonding. In the O_6_–H⋯O_3_ and N_1_–H⋯O_2_ interactions, the values found for the ratio |v|/G = 1.0, with h<0; however, these values do not indicate that the hydrogen bonds in these interactions have any covalent character, as the values of h(r) are minimal.

Closed-shell interactions are weak compared to shared interactions. The analysis of the NBO calculations showed that the hydrogen bonds in the supramolecular arrangements of GSH are stabilized by the hyperconjugation of lone pairs (Lewis type) with $\sigma^{*}$ antibonding orbitals (non-Lewis type) of the interacting region of the neighboring molecule at higher energies. The most significant results of the NBO calculations are found in Appendix A. They show that the higher hyperconjugation energies stabilize the molecules in the GSHB polymorph more than in the GSHA polymorph. As an example, we cite the interactions that are commonly found in both polymorphs with high $E^{\left(2\right)}$ energies. The O_6_–H⋯O_3_ interaction is stabilized by the hyperconjugation of the lone pairs of the O_3_ atom with the $\sigma^{*}$ antibonding orbitals of the O_6_–H bond, $\eta_{x}\left(O_{3}\right)\to \sigma^{*}\left(O_{6}-H\right)$, x=1;2. The hyperconjugation of the $\eta_{1}\left(O_{3}\right)$ orbital (occupancy 1.98*e*, *sp*^0.7^ hybrid) resulted in energy equal to 8.52 kcal/mol in GSHA and 7.80 kcal/mol in GSHB (ratio 1.1:1); on the other hand, the hyperconjugation of the $\eta_{2}\left(O_{3}\right)$ orbital (occupancy 1.88, pure *p*) resulted in energy equal to 9.45 kcal/mol in GSHA and 22.01 kcal/mol in GSHB (ratio 1:2.3). This means that the O_6_–H⋯O_3_ interaction is more cohesive in the GSHB polymorph. Two other essential interactions in both polymorphs were N_2_–H⋯O_2_ and N_3_–H⋯O_5_. The first is stabilized by the hyperconjugation of the two lone pairs of O_2_, $\eta_{1}$ (occupancy 1.98*e*, *sp*^0.6^ hybrid) and $\eta_{2}$ (occupancy 1.89*e*, pure *p*) with the $\sigma^{*}$ antibonding orbital of the N_2_–H bond in GSHA, with energies of 2.01 and 2.93 kcal/mol, but only by the lone pair $\eta_{2}$ (occupancy 1.96*e*, pure *p*) in GSHB, with an energy of 6.80 kcal/mol. In the N_3_–H⋯O_5_ interaction, only the $\eta_{2}$ lone pair of O_5_ (occupancy 1.86*e*, pure *p*) contributes to its stabilization in GSHA, while the two lone pairs of O_5_, $\eta_{1}$ (occupancy 1.97*e*, hybrid *sp*^0.7^) and $\eta_{2}$ (occupancy 1.87*e*, pure *p*) hyperconjugate with the $\sigma^{*}$ antibonding orbital of the N_3_–H bond in GSHB; however, both energies in this polymorph further stabilize the interaction.

### 2.4. Molecular Docking Analysis

The ACE (EC 3.4.15.1), a Zn metalloproteinase, plays an essential role in cardiovascular function by converting the decapeptide angiotensin I to angiotensin II (vasopressor octapeptide). There are two isoforms of ACE (somatic and testis) that are transcribed from the same gene in a tissue-specific manner [47]. The somatic form consists of two homologous domains (N and C domain), each containing an active site with a conserved zinc-binding motif. Despite the high sequence similarity between the two domains, they differ in substrate and inhibitor specificity and their activation by chloride ions [48]. The C domain seems to be the dominant angiotensin-converting site [47]. Docking calculations indicate that lisinopril had the higher affinity for ACE_C. [48]. Recently, the binding activity of several peptides with ACE inhibitory activity was modeled by in silico approaches, such as the quantitative structure–activity relationship (QSAR) and molecular docking. The methods provide details illuminating the interaction mechanism between the receptor and the ligands [48,49,50,51,52].

In the present work, using redocking analysis, molecular docking was validated for a co-crystallized ligand (lisinopril, a commercial ACE inhibitor) with the ACE. The same model employed in redocking lisinopril was also used for GSH. Seven of the ten poses presented RMSD values less than 2.0 Å. As shown in Figure 12, the residues involved in the interaction and the types of non-bonding interactions involved between the studied complexes were Gln281, Ala353, Ala354, Ala 356, Tyr520, Tyr523, His387, and Phe512 (Conventional hydrogen bond, Unfavorable Acceptor–Acceptor, Pi-Sulfur, van der Waals). Some compounds of the herbal species *Cucurbita pelo* L. presented potential interaction in silico with the ACE, and the His353 also interacted by a hydrogen bond. Thus, it seems that His353 represents a critical interaction point. The number and the distances of hydrogen bonds play an essential role in the potential biological interaction between the ligand and the binding site. According to Figure 13, the hydrogen bond distances ranged from 1.91 to 2.60 Å [53,54].

Furthermore, the same carboxyl group interacting with Gln281 also appears in the pharmacophore model (Figure 12c and Figure 14). GSH is a well-known endogenous antioxidant [55,56]. This substance is synthesized in the liver and is involved in various organic processes such as antioxidant defense, metabolism, and regulation. Moreover, after drug treatment, GSH reduces oxidative stress in the heart of diabetic rabbits and also has a protective effect in mice from sepsis by inhibiting the inflammation process [57,58]. Thus, GSH may have protective cardiovascular effects associated with its antioxidant properties, which could provide vascular protection. Furthermore, GSH presents a potential impact on antihypertension, and its antioxidant properties can reduce vascular damage [59].

In addition, when oxidative stress creates vascular damage by promoting cell growth, extracellular matrix protein deposition, endothelial dysfunction, and increased vascular tone, these features contribute to the vascular phenotype in hypertension [60]. In the study performed by Bessa, S. S. and colleagues (2009), it was observed that in hypertensives, both systolic and diastolic pressures were negatively correlated with glutathione S-transferase in Egyptian patients [61]. Beyond oxidative stress, our results suggest that glutathione might interact with some regulatory points in the cardiovascular system, such as the angiotensin-converting enzyme, playing an essential role in blood pressure regulation.

## 3. Computational Procedures

### 3.1. Computational Methods

The crystallographic information files of GSHA and GSHB were obtained from the Cambridge Crystallographic Data Centre (CCDC) [62] under codes 762195 and 1500579, respectively. Theoretical calculations were carried out for both conformations [63] using the highly parameterized empirical exchange–correlation functional M06-2X [31], combined with the triple-ζ basis set 6-311++G(d,p), in gas phase, implemented in the Gaussian16 [64] software package. Frontier molecular orbitals (FMOs) [63], the highest occupied molecular orbital (HOMO), and the lowest unoccupied molecular orbital (LUMO) were calculated and the chemical reactivity and kinetic stability of both molecules were obtained. To verify the electronic charge distribution on the GSH molecular surface, the MEP [65] map was obtained through electrostatic potential V(r) [66],
(8)$$V\left(r\right)=\overset{}{\underset{\alpha }{\sum }}\frac{Z_{\alpha }}{\left|r_{\alpha }-r\right|}-\int \frac{\rho \left(r^{\prime }\right)}{\left|r^{\prime }-r\right|}dr^{\prime },$$
where $Z_{\alpha }$ is the charge of nuclei α at point $r_{\alpha }$ and $\rho \left(r^{\prime }\right)$ is the charge density at the point $r^{\prime }$. This function has been used for predicting nucleophilic and electrophilic regions and is valuable in studying intermolecular interactions. To predict the local electrophilicity, we used the Fukui function [67]
(9)$$f\left(r\right)={\left[\frac{\partial \rho \left(r\right)}{\partial N}\right]}_{v},$$
where N is the number of electrons present in the system and the constant term v in the partial derivative is external potential. Fukui indices have been widely used to predict the reactive sites of organic molecules, so the function values are higher in these regions than in regions with no reactivity. All calculations were carried out by DFT, implemented in the Gaussian 16 [64] software package at the M06-2X/6-311++G(d,p) level of theory in the gas phase, and the results were obtained by the GaussView 6 [68] software.

### 3.2. Supramolecular Arrangement

HS analysis using the CrystalExplorer program [69] was employed to study the intermolecular interactions [39]. The normalized distance ($d_{norm}$) is constructed by the distance from the surface to the nearest nucleus inside the surface ($d_{i}$) and outside the surface ($d_{e}$), the distance from the surface to the nearest inner atom, and the van der Waals radii of the internal and external atoms ($r_{i}^{vdW}$ and $r_{e}^{vdW}$) [39],
(10)$$d_{norm}=\frac{d_{i}-r_{i}^{vdW}}{r_{i}^{vdW}}+\frac{d_{e}-r_{e}^{vdW}}{r_{e}^{vdW}}\;$$

In addition, an analysis of intermolecular contacts and their contributions to the packaging of crystals, based on the combination of $d_{e}$ vs. $d_{i}$ plots, generate fingerprints, which summarize the percentage contribution to the nature and type of intermolecular interaction present in the molecule [41]. The topological parameters of GSH molecular systems were obtained by the QTAIM [43] by Multiwfn software [70], and the stabilities of the interaction were verified by NBO [71] calculations; a second-order perturbation theory formula provides the hyperconjugation energies:(11)$$E_{i\to j^{*}}^{\left(2\right)}=-n_{\sigma }\frac{<\sigma_{i}|\hat{F}|\sigma_{j}^{*}>{}^{2}}{\varepsilon_{j^{*}}-\varepsilon_{i}}=-n_{\sigma }\frac{F_{ij}^{2}}{\varepsilon_{j^{*}}-\varepsilon_{i}},$$
where $<\sigma |F|\sigma {>}^{2}$ or $F_{ij}^{2}$ is the Fock matrix element between the natural bond orbitals i and *j*; $\varepsilon_{\sigma^{*}}$ is the energy of the antibonding orbital $\sigma^{*}$ and $\varepsilon_{\sigma }$ is the energy of the bonding orbital σ; and $n_{\sigma }$ stands for the population occupation of the σ donor orbital. Theoretical calculations were carried out at the M06-2X/6-311G++(d,p) level of theory, in which the atom positions were fixed in their crystallographic positions.

### 3.3. Molecular Docking Analysis

ACE (PDB ID: 1O86) was downloaded from Protein Data Bank (https://www.rcsb.org; accessed on 12 September 2022) for molecular docking simulations. The enzyme was prepared by adding hydrogen atoms, and water molecules were removed. Next, the GSH molecule was saved in mol2 format. After that, the binding site of the macromolecule was selected, and the grid box was determined (X, Y, and Z coordinates). The downloaded file was saved in PDB format before being subjected to GOLD Suite 5.7.0 (Mark Thompson and Planaria Software LLC) [72,73] to locate the binding site and determine the binding affinity. Biovia Discovery Studio 3.5 software was employed to obtain the 2D interaction figures, and PyMOL Molecular Graphics System 2.0 was used to obtain the 3D interaction images. Redocking with the co-crystallized ligand (lisinopril) was carried out to validate the models obtained. The CHEMPLP score function was used.

### 3.4. Pharmacophore Design

Binding DB (https://www.bindingdb.org/bind/index.jsp; accessed on 7 September 2022) was used to identify the compounds with antagonist activity against the ACE. The five molecules with the lowest half-maximal inhibitory concentration (IC_50_) were selected to obtain pharmacophore models. PharmaGist web server [73,74] (https://bioinfo3d.cs.tau.ac.il/PharmaGist/; accessed on 10 September 2022) was employed to obtain the 3D pharmacophore model from the set of ACE antagonists and GSH. The algorithm determines potential pharmacophores and selects the highest-scoring ones. The suggested pharmacophore hits were determined after the spatial alignment of the GSH with the five EGFR antagonists. Pharmacophore analysis was used to search the possible critical features of ACE antagonists responsible for the interaction of the binding site. Hydrogen bond acceptors, donors, hydrophobic groups, and aromatic rings were selected to generate the pharmacophore models. A minimum of three to a maximum of six elements was considered. The following weights were used as default in the program (3.0 for aromatic rings, 1.0 for charges, 1.5 for hydrogen bond donors or acceptors, and 0.3 for hydrophobic groups). The images were generated in Biovia Discovery Studio 3.5

## 4. Conclusions

A comparison of the crystalline structures of GSH showed that the torsion of the carboxyl group in the glycine portion did not change the crystalline characteristics of the polymorphs, so they are in the same space group. Furthermore, the theoretical calculations showed that, when kept isolated, the molecule tends to remain in the neutral form, while in the crystalline environment, the zwitterionic form is predominant. The calculations also proved the antioxidant power of GSH by the frontier molecular orbitals and showed the regions where electrophilic and radical attacks occur in chemical processes. In supramolecular arrangements, some variations were observed in the contacts that form the intermolecular interactions, so they are more stabilized in GSHB compared to GSHA. In addition, the molecular docking analysis showed interactions between the active site residues of the ACE and glutathione through seven hydrogens. Moreover, these hydrogen bond acceptors are essential for binding the GSH with the ACE. The observed results suggest that glutathione analogs would be attractive dual-acting (antioxidant plus ACE inhibitor) antihypertensive agents and provide useful information for developing GSH analogs with higher ACE inhibitor activity.

## Figures and Tables

**Figure 1 molecules-27-07958-f001:**
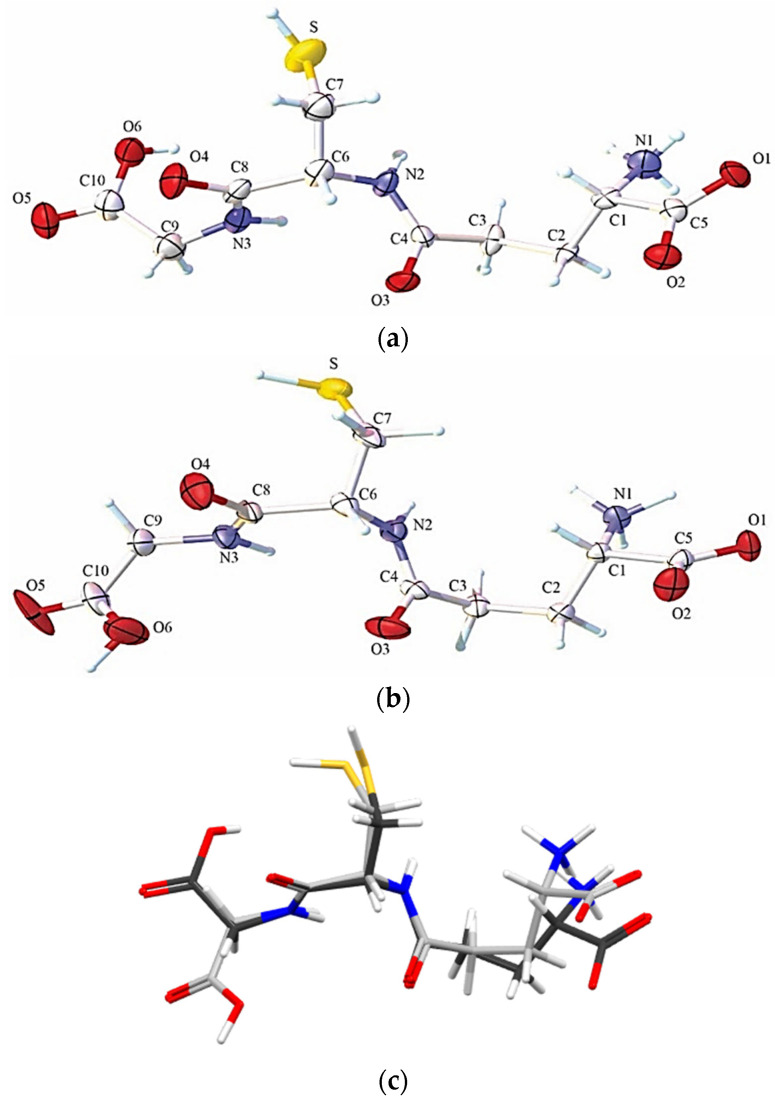
ORTEP diagram for (**a**) GSHA, (**b**) GSHB, and (**c**) the overlapping structure. The ellipsoids are represented at a 75% probability level with the atomic numbering scheme. The hydrogen atoms are represented by spheres with arbitrary radii. (**c**) The overlap of GSHA and GSHB polymorphs.

**Figure 2 molecules-27-07958-f002:**
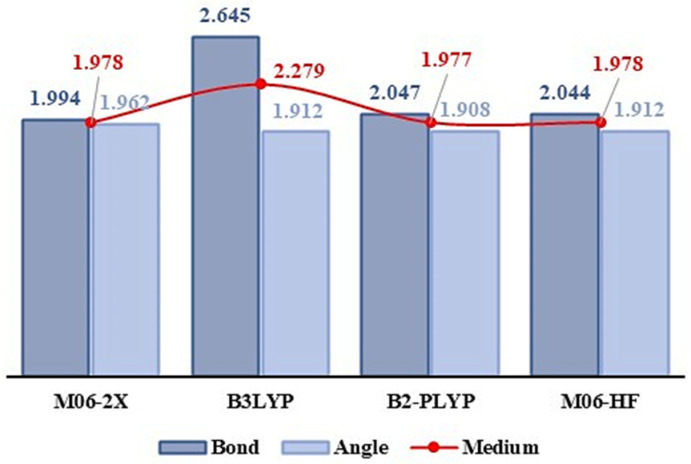
Graph of the mean absolute deviations for geometric parameters obtained by the exchange and correlation functionals M06-2X, B3LYP, B2-PLYP, and M06-HF (compared to experimental results).

**Figure 3 molecules-27-07958-f003:**
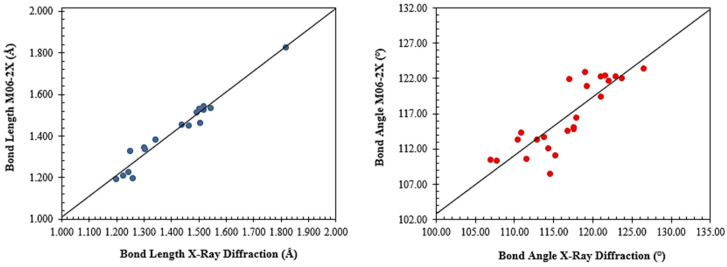
Comparison graphs of the geometric parameters’ bond length and angle, obtained by XRD and the M06-2X/6-311++G(d,p) level of theory for the glutathione.

**Figure 4 molecules-27-07958-f004:**
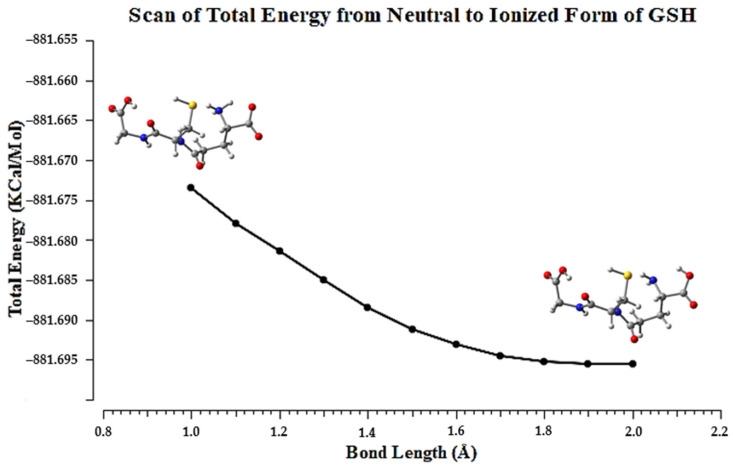
Plot showing the total energy derived from a relaxed potential energy surface sweep for the proton transition to the neutral and ionized forms of GSH, varying the distance N_1_⋯H by 1.0 Å in 0.1 Å increments.

**Figure 5 molecules-27-07958-f005:**
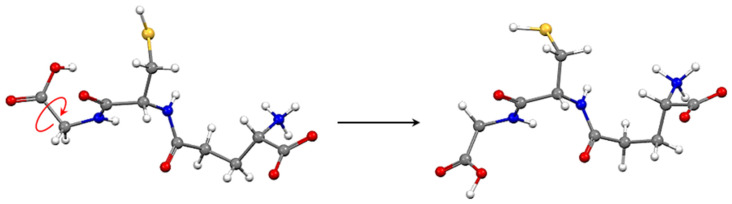
Torsion of the carboxyl group, -COOH, of the glycine portion in the glutathione molecule, and the overlapping of their molecular structures from the peptide bond that joins the glycine chain to the cysteine chain.

**Figure 6 molecules-27-07958-f006:**
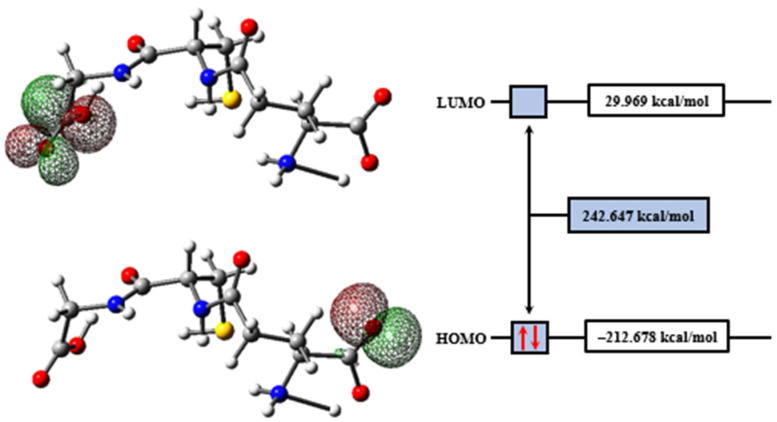
HOMO and LUMO plots for GSH calculated at the M06-2X/6-311++G(d,p) level of theory.

**Figure 7 molecules-27-07958-f007:**
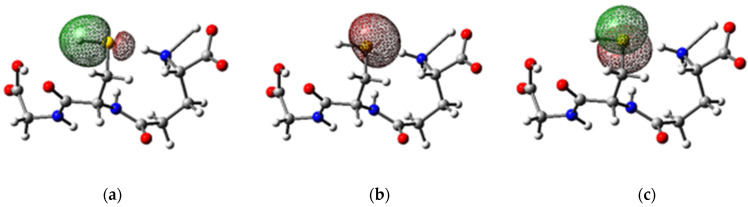
Isosurface plots of the (**a**) s orbitals of the S-H bond, and of the lone pairs (**b**) $\eta_{1}$(S) and (**c**) $\eta_{2}$ (S) of the S atom in glutathione, calculated at the M06-2X/6-311++G(d,p) level of theory.

**Figure 8 molecules-27-07958-f008:**
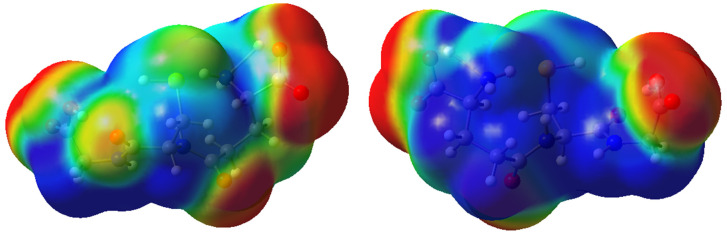
MEP surface at *ρ*(r) = 4.0 × 10^−4^ electrons/Bohr^3^ contour of the total SCF electronic density for glutathione molecule at the M06-2X/6-311++G(d,p) level of theory.

**Figure 9 molecules-27-07958-f009:**
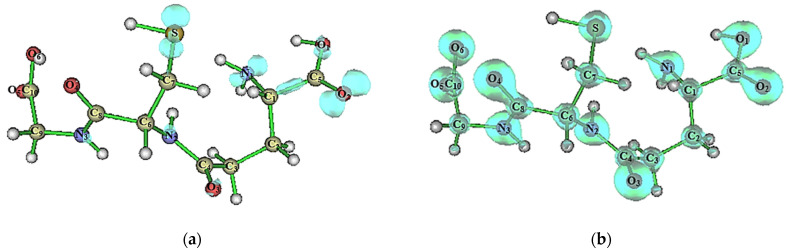
Isosurfaces of the (**a**) $f^{0}$ and (**b**) $f^{-}$ functions, calculated at a proper value of 0.5 for the glutathione molecule.

**Figure 10 molecules-27-07958-f010:**
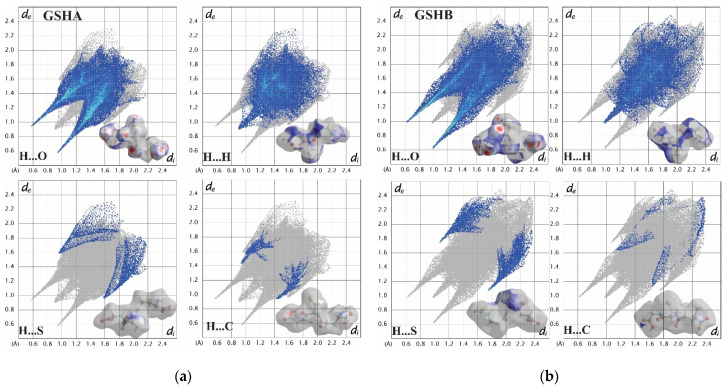
Fingerprint plots for (**a**) GSHA and (**b**) GSHB.

**Figure 11 molecules-27-07958-f011:**
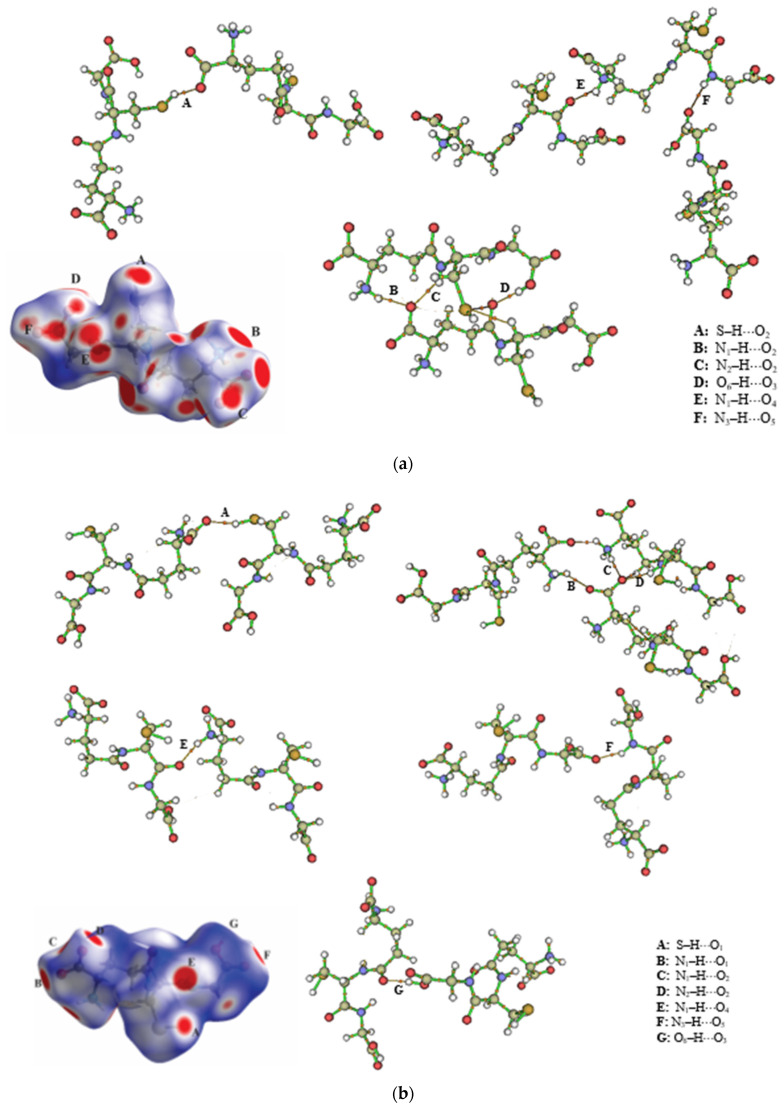
Molecular graphs of some bond paths of the intermolecular interactions in the supramolecular arrangements of the (**a**) GSHA and (**b**) GSHB polymorphs.

**Figure 12 molecules-27-07958-f012:**
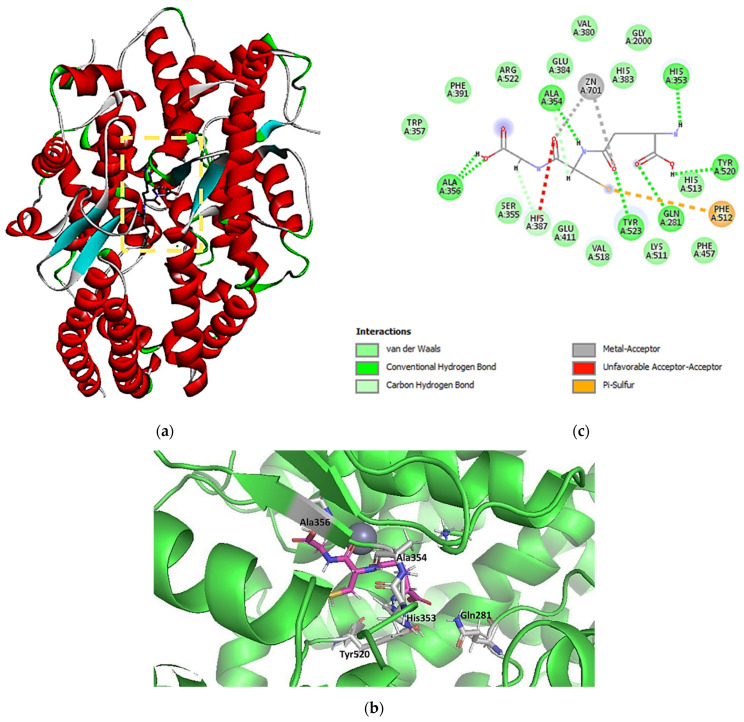
Crystal structure of lisinopril complexed with lisinopril (**a**); 2D (**b**) and 3D (**c**) forms of interaction between glutathione with the ACE binding site (pose 1; PDB ID: 1O86).

**Figure 13 molecules-27-07958-f013:**
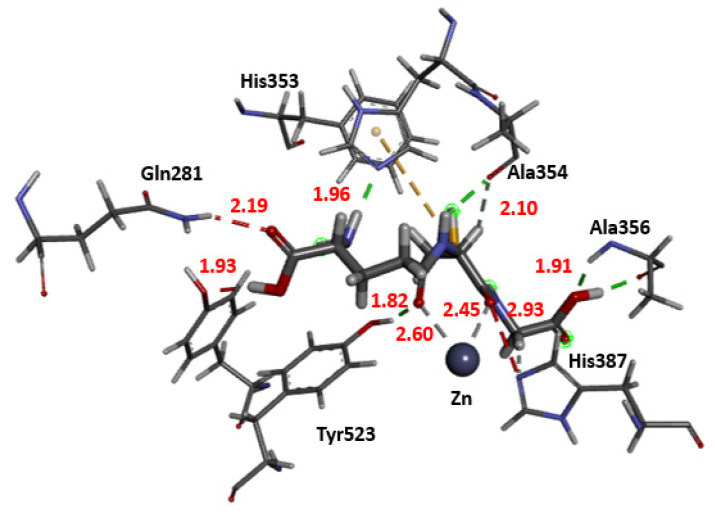
3D representation of distance interactions of the glutathione and angiotensin-converting enzyme binding site.

**Figure 14 molecules-27-07958-f014:**
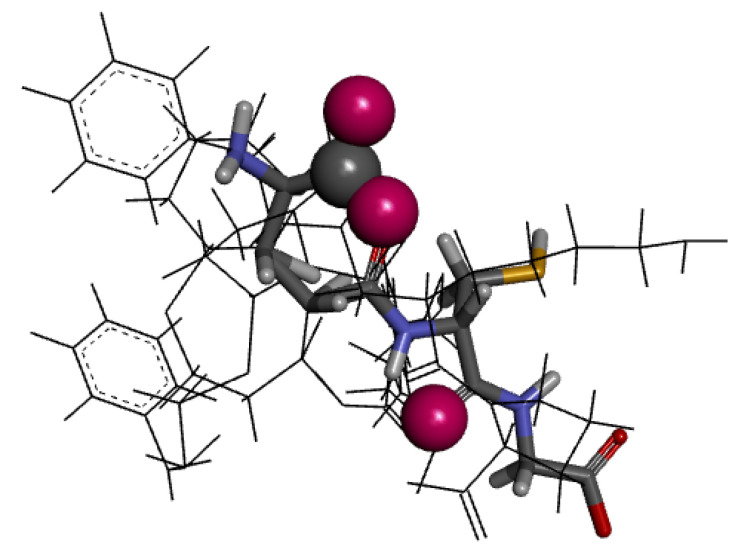
Spatial fitting of glutathione to the angiotensin-converting enzyme (ACE) antagonist pharmacophore. Hydrogen-bond acceptors and negative pharmacophore features are represented in red and gray, respectively.

**Table 1 molecules-27-07958-t001:** Crystallographic data and structure refinement parameters for GSHA and GSHB.

Crystallographic Data	GSHA	GSHB
Empirical formula	C_10_H_17_N_3_O_6_S	C_10_H_17_N_3_O_6_S
Formula weight	307.33	307.33
Crystal system	orthorhombic	orthorhombic
Space group	P2_1_2_1_2_1_	P2_1_2_1_2_1_
a (Å)	5.2748(2)	5.6131(11)
b (Å)	8.3459(3)	8.720(2)
c (Å)	25.496(3)	27.940(5)
α (°)	90	90
β (°)	90	90
γ (°)	90	90
Volume (Å^3^)	1122.39(14)	1367.6(5)
Z	4	4
ρ_calc_ (g/cm^3^)	1.819	1.492
μ (mm^−1^)	0.077	0.146
F(000)	648.0	648.0
Crystal size/mm^3^	0.2 × 0.2 × 0.1	0.19 × 0.14 × 0.11
Radiation	synchrotron (λ = 0.47670)	synchrotron (λ = 0.5636)
2Θ range for data collection/°	3.444 to 34.714	2.312 to 55.978
Index ranges	−6 ≤ *h* ≤ 6 −10 ≤ *k* ≤ 10 −14 ≤ *l* ≤ 14	−9 ≤ *h* ≤ 9 −14 ≤ *k* ≤ 14 −46 ≤ *l* ≤ 44
Reflections collected	5512	36,183
Goodness of fit on F^2^	1.115	1.268
Final R indexes [all data]	R_1_ = 0.0483, wR_2_ = 0.0449	R_1_ = 0.0340, wR_2_ = 0.0780
Largest diff. peak/hole (eÅ^−3^)	0.24/−0.29	0.50/−0.53
Flack parameter	−0.2(3)	−0.01(4)

**Table 2 molecules-27-07958-t002:** Glutathione thermochemical properties obtained at the M06-2X/6-311++G(d,p) level of theory.

Thermochemical Property	Neutral	Zwitterion	Diff *
Electronic Energy (kcal/mol)	−881,611.16	−881,477.12	−134.03
Zero-Point Energy (kcal/mol)	−881,430.11	−881,287.93	−142.18
Internal Energy (kcal/mol)	−881,419.18	−881,280.20	−138.98
Enthalpy (kcal/mol)	−881,418.59	−881,279.61	−138.98
Free Energy (kcal/mol)	−881,458.22	−881,313.73	−144.48
Entropy (cal/molK)	132.92	114.45	18,47
Heat Capacity (cal/molK)	69.59	50.57	-
Polarizability (a.u.)	174.96	162.49	-
Dipole Moment (Debye)	6.46	13.45	-

$*\;Diff=\varepsilon_{n}-\varepsilon_{z}$, where $\varepsilon_{n}$ is the thermochemical property of the neutral form and $\varepsilon_{z}$ is the ionized form.

**Table 3 molecules-27-07958-t003:** Reactivity indices for the glutathione obtained at the M06-2X/6-311++G(d,p) level of theory.

Descriptors	(kcal/mol)
$E_{\mathrm{HOMO}}$	−212.678
$E_{\mathrm{LUMO}}$	29.969
GAP	242.647
Ionization Energy (I)	212.678
Electronic Affinity (A)	−29.969
Electronegativity (χ)	91.355
Chemical Potential (μ)	−91.355
Chemical Hardness (η)	242.647
Chemical Softness (σ)	0.004
Electrophilicity Index (ω)	17.197

**Table 4 molecules-27-07958-t004:** Hydrogen bond geometry and topological parameters by QTAIM obtained for GSHA and GSHB glutathione molecular interactions.

D–H⋯A	D–H (Å)	H⋯A (Å)	D⋯A (Å)	D–H⋯A (°)	$\rho_{BCP}$^(*a*)^ (a.u.)	$\nabla^{2}\rho_{BCP}$^(*b*)^ (a.u.)	G(r)^(*c*)^ (a.u.)	v(r)^(*d*)^ (a.u.)	h(r)^(*e*)^ (a.u.)	$\frac{\left|v\left(r\right)\right|}{G\left(r\right)}$
GSHA										
S–H⋯O_1_ ^I^	1.2000	2.5900	3.691(5)	152	0.0175	0.0655	0.0140	−0.0116	0.0024	0.8
S–H⋯O_2_ ^I^	1.2000	2.1500	3.261(8)	153′	0.0174	0.0660	0.0140	−0.0115	0.0025	0.8
O_6_–H⋯O_3_ ^II^	0.8300	1.6600	2.465(4)	164	0.0469	0.2126	0.0543	−0.0554	−0.0011	1.0
N_1_–H⋯O_1_ ^III^	0.8900	1.8200	2.669(10)	157′	0.0134	0.0512	0.0110	−0.0091	0.0018	0.8
N_1_–H⋯O_2_ ^II^	0.8900	2.2900	2.947(4)	130	0.0302	0.1483	0.0331	−0.0291	0.0040	0.9
N_1_–H⋯O_4_ ^IV^	0.8900	2.4800	3.122(6)	130	0.0085	0.0319	0.0067	−0.0055	0.0012	0.8
N_2_–H⋯O_2_ ^II^	0.8600	1.9900	2.675(7)	136′	0.0244	0.1175	0.0251	−0.0208	0.0043	0.8
N_3_–H⋯O_5_ ^V^	0.8600	2.0700	2.672(8)	127′	0.0214	0.1034	0.0216	−0.0173	0.0043	0.8
C_2_–H⋯O_4_ ^IV^	0.9700	2.4200	3.097(5)	127	0.0116	0.0416	0.0089	−0.0075	0.0015	0.8
GSHB										
S–H⋯O_1_ ^VII^	1.3400	2.1800	3.4603(8)	158	0.0154	0.0538	0.0114	−0.0094	0.0020	0.8
N_1_–H⋯O_4_ ^VIII^	1.0200	1.8400	2.8034(6)	155	0.0310	0.1209	0.0285	−0.0267	0.0018	0.9
N_1_–H⋯O_1_ ^IX^	1.0200	1.9600	2.8383(7)	142′	0.0222	0.0956	0.0208	−0.0177	0.0031	0.9
N_1_–H⋯O_2_ ^II^	1.0200	1.7200	2.6959(6)	158′	0.0438	0,1527	0.0407	−0.0431	−0.0025	1.1
N_2_–H⋯O_2_ ^II^	1.0100	1.9600	2.8984(7)	154	0.0222	0.0993	0.0214	−0.0179	0.0035	0.8
N_3_–H⋯O_5_ ^X^	1.0100	2.0000	2.8712(7)	144′	0.0214	0.0924	0.0199	−0.0166	0.0032	0.8
O_6_–H⋯O_3_ ^XI^	0.9600	1.6400	2.5987(6)	172	0.0495	0.1628	0.0463	−0.0519	−0.0056	1.1
C_3_–H⋯O_2_ ^II^	1.0900	2.5200	3.3842(8)	135	0.0098	0.0297	0.0068	−0.0062	0.0006	0.9
Symmetry codes:		(I) $\frac{3}{2}+x,\;\frac{3}{2}-y,\;1-z$		(V) $2-x,\;-\frac{1}{2}+y,\;\frac{1}{2}-z$		(IX) $\frac{1}{2}+x,\;\frac{3}{2}-y,\;-z$				
		(II) 1+x, y, z		(VI) −1+x, y, z		(X) $2-x,\;\frac{1}{2}+y,\;\frac{1}{2}-z$				
		(III) $\frac{1}{2}+x,\;\frac{1}{2}-y,\;1-z$		(VII) 1+x, −1+y, z		(XI) $1-x,\;-\frac{1}{2}+y,\;\frac{1}{2}-z$				
		(IV) −1+x, −1+y, z		(VIII) x, 1+y, z						
Topological Properties:		^(*a*)^ Total electronic density on BCP		^(*c*)^ Lagrangian Kinetic energy		^(*e*)^ Total energy density				
		^(*b*)^ Laplacian of electron density on BCP		^(*d*)^ Potential energy density						

## Data Availability

The data presented in this study are available in Appendix A.

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
