# Peer review of "New Insights on Glutathione’s Supramolecular Arrangement and Its In Silico Analysis as an Angiotensin-Converting Enzyme Inhibitor"

_molecules, 2022, doi:10.3390/molecules27227958_

Round 1

Reviewer 1 Report

Since the results of in vitro assay for ACE inhibitory activity using the glutathione depending structural change are not supported in this paper, the result of in silico analysis is not validated by ACE inhibitory assay. Authors need to suggest in vitro assay data to validate and compare to ACE inhibitors such as lisinopril as control. 

Author Response

O levantamento da literatura mostrou atividade inibidora da ECA comparativa in vitro de GSH (Referências 1 e 2) e lisinopril (Referência 2). Os dados demonstraram a propriedade inibidora da ECA de GSH. Nesse sentido, o texto foi devidamente modificado, referindo-se às atividades inibidoras comprovadas experimentalmente.

Referências:

  1. Zehra Basi, Efeito de inibição da nicotinamida (vitamina B3) e peptídeo de glutationa reduzida (GSH) na atividade da enzima conversora de angiotensina purificada de rim de ovelha. Int J Biol Macromol. 2021; 189, 65-71. doi: 10.1016/j.ijbiomac.2021.08.109.

IC50 values for nicotinamide and GSH were calculated as 14.3 μM and 7.3 μM, respectively. Type of inhibition and Ki values for nicotinamide and GSH from the Lineweaver-Burk graph were determined. The type of inhibition for nicotinamide and GSH was determined as non-competitive inhibition. Ki value was calculated as 15.4 μM for nicotinamide and 6.7 μM for GSH.

  1. ZehraBasi, VedatTurkoglu, In vitro effect of oxidized and reduced glutathione peptides on angiotensin converting enzyme purified from human plasma. Journal of Chromatography B, Volume 1104, 1 January 2019, Pages 190-195. https://doi.org/10.1016/j.jchromb.2018.11.023

Os valores de IC50 e Ki para o peptídeo GSH foram calculados para 16,2 μM e 11,7 μM, respectivamente. Os valores de IC50 para GSH e lisinopril foram determinados em 16,2 μM e 0,781 nM, respectivamente. O tipo de inibição para GSH e lisinopril do gráfico Lineweaver-Burk foi uma inibição reversível não competitiva e as constantes Ki para GSH e lisinopril foram calculadas como 11,7 μM e 0,662 nM, respectivamente . ”

Reviewer 2 Report

Authors presented properties like Electrophilicity, hardness, Docking analysis of ACE and glutathione using DFT studies.  These are all very useful for the new experiments.

Authors should have tested different DFT methods and basis sets at least 3-4 methods with one good basis set.

Author Response

The level of theory initially used in this work was validated based on crystallographic data. However, we carried out new calculations with the exchange-correlation functionals B3LYP, B2-PLYP, and M06-HF, in which we confirmed the efficiency of the functional M06-2X to describe the molecular systems. The triple-zeta basis set 6-311++G(d,p) was kept in all calculations.
